# Universal Deformations

**Aleksey Cherman** [⋆], **Theodore Jacobson** [†] **and Maria Neuzil** [‡]

School of Physics and Astronomy, University of Minnesota, Minneapolis MN 55455, USA

⋆ acherman@umn.edu, † jaco2585@umn.edu, ‡ neuzi008@umn.edu

## Abstract

QFTs with local topological operators feature unusual sectors called "universes," which are separated by infinite-tension domain walls. We show that such systems have relevant deformations with exactly-calculable effects. These deformations allow one to dial the vacuum energy densities of the universes. We describe applications of these deformations to confinement in 2d gauge theories, as well as a curious violation of the effective field theory naturalness principle.



# 1  Introduction and summary

In recent years it has become appreciated that one can productively reinterpret the study of symmetries in quantum field theory as the study of various topological operators [1]. For example, if a QFT in $d$ Euclidean spacetime dimensions has a conventional $U(1)$ global symmetry, there is a conserved 1-form current $j = j_\mu dx^\mu$, and one can define the operator

$$U_\alpha(M_{d-1}) = \exp\left( i\alpha \int_{M_{d-1}} \star j \right), \tag{1}$$

where $M_{d-1}$ is a closed $(d-1)$-dimensional manifold, and $\alpha \equiv \alpha + 2\pi$ because we assume that $j$ is normalized such that $\int_{M_{d-1}} \star j \in \mathbb{Z}$. The operator $U_\alpha(M_{d-1})$ is topological: conservation of $j$ implies that when $U_\alpha(M_{d-1})$ is inserted into any correlation function, one can freely deform $M_{d-1}$ without changing the correlator, so long as $M_{d-1}$ does not cross certain local operators. Such local operators are precisely those that create the particles charged under the $U(1)$ symmetry, and $U_\alpha(M_{d-1})$ can be thought of as a generator of the symmetry. The fact that $U_\alpha(M_{d-1})$ generates a $U(1)$ symmetry is encoded in the "fusion rule" of two operators defined on the same codimension-1 manifold,

$$U_\alpha(M_{d-1})U_\beta(M_{d-1}) = U_{\alpha+\beta}(M_{d-1}). \tag{2}$$

An operator with charge $q$ under the $U(1)$ symmetry obeys

$$U_\alpha(M_{d-1})\mathcal{O}_q(x) = e^{iq\alpha \ell(M_{d-1},x)} U_\alpha(\widetilde{M}_{d-1})\mathcal{O}_q(x), \tag{3}$$

where the integer $\ell(M_{d-1}, x)$ is the linking number of $M_{d-1}$ and $x$, and $\widetilde{M}_{d-1}$ is a smooth deformation of $M_{d-1}$ which has zero linking number with $x$.

The above perspective is useful because it enables many generalizations. If one views a symmetry as being *defined* by the existence of a set of topological operators with some fusion rules like Eq. (2) and "commutation" rules like Eq. (3), then:

- One can discuss discrete symmetries on the same footing as continuous symmetries, even though generators of discrete symmetries often cannot be written in terms of an integral of a local conserved current.

- One can consider symmetries that act on extended objects like strings and membranes. Symmetries that act on operators defined on $n$-dimensional closed manifolds are generated by $(d-n-1)$-dimensional topological operators, and are called $n$-form symmetries.

- One can discuss symmetries that are not visible at the level of a Lagrangian description of a theory.

- Finally, one can study symmetries generated by a set of topological operators $U_\alpha(M)$ that satisfy rather general fusion rules of the form[1]

  $$U_\alpha(M)U_\beta(M) = \sum_\gamma N_{\alpha\beta}^\gamma U_\gamma(M). \tag{4}$$

  If there is more than one term on the right, then the symmetry generators $U_\alpha(M)$ do not generate a symmetry group. Instead, they can generate a symmetry category, see e.g. Refs. [2–23]. In particular, studying QFTs with topological operators allows one to discuss QFTs with $n$-form symmetries where some symmetry generators do not have an inverse.

---

[1]We assume that the set of topological operators does not include the zero operator, and that for each pair of $U_\alpha, U_\beta$ there exists a $U_\gamma$ such that $N_{\alpha\beta}^\gamma \neq 0$.

Here we will study QFTs with local topological operators $U_\alpha(x)$, where $x$ is a point on the spacetime manifold, see e.g. [9, 18, 19, 24–42]. These operators generate a $(d-1)$-form symmetry. Such QFTs are especially ubiquitous in two spacetime dimensions, and include e.g. charge-$N$ 2d QED and pure $SU(N)$ 2d Yang-Mills theory. Some examples are also known in $d > 2$, see e.g. Refs. [32, 39]. We will show that QFTs with local topological operators always have some relevant deformations. These deformations are produced by adding spacetime integrals of local topological operators to the action

$$\Delta_\alpha S = \int_{M_d} d^d x \, \tfrac{1}{2} \left( \Lambda_\alpha^d U_\alpha(x) + \text{h.c.} \right), \tag{5}$$

where $\Lambda_\alpha$ is the mass scale of the deformation and $M_d$ is the spacetime manifold. These perturbations of the action have a rare and luxurious property: their effects are exactly calculable.

Perturbations like Eq. (5) are dimension-0 deformations. They share this property with the cosmological constant. However, dialing the cosmological constant in a QFT without a coupling to gravity does not affect any of its correlation functions. In contrast, we will see that dialing $\Lambda_\alpha$ affects the correlation functions of $(d-1)$-dimensional operators (e.g. line operators in $d = 2$), and so the deformations we consider affect the phase structure of the QFTs that we study.

The way this happens is as follows. QFTs with local topological operators (LTOs) have sectors that are labeled by their expectation values. A domain wall which connects any two vacua belonging to sectors with different expectation values of LTOs must have infinite tension. Heuristically, if a finite-tension domain wall exists, then the expectation values of local operators would evolve smoothly through the wall. But this is impossible for LTOs because they are topological — their expectation values cannot change continuously as a function of spacetime. The infinite domain wall tension means that in contrast to conventional vacua in QFT, vacua (and states) that are labeled by distinct LTO expectation values cannot mix even in finite volume. This motivates referring to the vacua labeled by LTOs as belonging to distinct "universes." This helpfully-evocative term was first introduced in Ref. [25].

The fact that it is impossible for local excitations in one universes to "see" other universes makes one wonder whether it might be possible to dial the vacuum energy density of each universe separately from the others by dialing the coefficients of some local operators in the action. We will see that this is precisely the effect of the relevant deformations like Eq. (5), as illustrated in Fig 1. The basic idea is that the LTOs are constant within each universe, so one can replace $U_\alpha$ with its expectation value in the action. We call LTO deformations "universal deformations" because first, they have an exactly-calculable and universal effect in any QFT with a $(d-1)$ symmetry; and second, this effect is simply to shift the relative vacuum energies of the universes.

Local topological operators generate symmetries which act on $(d-1)$-dimensional operators. This is encoded in an expression akin to Eq. (3),

$$U_\alpha(x) W_q(C_{d-1}) = \gamma(\alpha, q, \ell(x, C_{d-1})) U_\alpha(\tilde{x}) W_q(C_{d-1}), \tag{6}$$

where $W_q(C_{d-1})$ is a charged operator defined on a $(d-1)$-dimensional closed manifold $C_{d-1}$, $q$ is a label encoding the charge of $W$ (which could be an integer or something more complicated, like a representation of some non-abelian group), the form of $\gamma$ depends on the example, and we assume $\ell(\tilde{x}, C_{d-1}) = 0$. If $C_{d-1}$ is sufficiently large and we imagine using the topological property of the $U_\alpha$ operators to move them far from the operators $W_q(C_{d-1})$ and invoke cluster decomposition, we learn that Eq. (6) implies that the expectation value of $U_\alpha(x)$ jumps when crossing $W_q(C_{d-1})$. This means that domain walls between universes can be interpreted as world-volumes of certain probe excitations — precisely the excitations whose absence gives

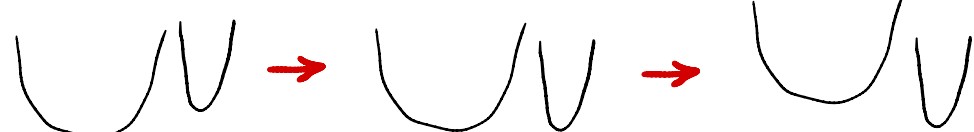

Figure 1: Each sketch above shows the effective potential energies in two universes as a function of the vacuum expectation value of some scalar field. The three sketches are related by dialing the coefficient of an appropriate "universal deformation" by a local topological operator. On the left, the deformation is turned off, and the two universes have distinct vacuum energies. If we dial the coefficient of the deformation, we can make their vacuum energies coincide (middle figure), or change which universe has the lower vacuum energy density (right figure).

rise to the symmetry generated by the LTOs. An immediate consequence is that the comparative vacuum energy densities of the universes determine whether these probes are confined, as illustrated in Figure 2.

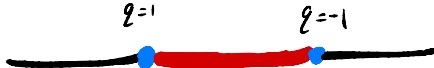

Figure 2: Two static test particles with charge ±1 are held at a fixed large distance from each other in a 2d abelian gauge theory where the dynamical fields have charge $N > 1$. If the universe in between the particles (shown in red at a fixed time) has a higher vacuum energy density than the universe outside, the particles will be confined by a linear potential.

The fact that our universal deformations affect the vacuum energy densities within individual universes implies that they also affect confinement of probe excitations. Since (de)confinement of probes is related to the realization of the symmetry generated by the LTOs, our universal deformations affect the phase structure of QFTs with LTOs.

In Sec. 2 we explain the above statements for invertible symmetries in a general setting. Section 3 contains a discussion of universal deformations in a specific instructive example: charge-$N$ QED in two dimensions (the charge-$N$ Schwinger model). Then in Sec. 4 we apply our general result to non-invertible symmetries, and explain its implementation in 2d pure $SU(N)$ Yang-Mills (YM) theory. We discuss some implications of our results in Sec. 5, where we comment on the status of a Coleman-Mermin-Wagner-type theorem for generalized symmetries, and highlight a startling violation of the Effective Field Theory (EFT) naturalness principle implied by our results.

Finally, in Appendix A we explain how to construct explicit expressions for topological operators in terms of fields in representative examples including the 2d compact scalar and 4d Maxwell QFTs. The discussion is aimed at clarifying a subtlety which has not been highlighted in the literature to date.

## 2 Deformation by invertible local topological operators

A set of invertible local topological operators generates an abelian $(d-1)$-form global symmetry. Without any appreciable loss of generality, in this section we can consider a discrete $\mathbb{Z}_N$ $(d-1)$-form global symmetry, for which the set of LTOs is finite. The statement that a QFT has a $(d-1)$-form $\mathbb{Z}_N$ global symmetry means that it has a set of $N$ topological local operators with elements $U_n(x)$ which obey

$$U_m(x)U_n(x) = U_{m+n \, \mathrm{mod}\, N}(x), \qquad m, n = 0, 1, \ldots, N-1. \tag{7}$$

One of these operators, say $U_0$, is simply the trivial unit operator, but the rest are more interesting. We also assume that there exist charge $q$ operators $W_q(C_{d-1})$ defined on closed manifolds $C_{d-1}$, so that Eq. (6) becomes

$$U_n(x)W_q(C_{d-1}) = \exp\left(iq\frac{2\pi n}{N}\ell(x, C_{d-1})\right)U_n(\tilde{x})W_q(C_{d-1}). \tag{8}$$

The statement that the operators $U_n(x)$ are topological means that

$$\langle U_{n_1}(x_1)U_{n_2}(x_2)\cdots\rangle \tag{9}$$

does not depend on the insertion positions $x_1, x_2, \cdots$. Moreover, if we note the $U_n^\dagger(x) = U_{N-n \, \mathrm{mod}\, N}(x) = U_n^{-1}(x)$, the relations above imply

$$\langle U_n^\dagger(x)U_n(0)\rangle = 1 \tag{10}$$

for any $x$. This is because we are free to move $x$ to $0$ without changing the correlator, and $U_n^\dagger(0)U_n(0)$ is the identity operator.

Systems with the $\mathbb{Z}_N$ LTOs descibed above (and no other LTOs) have $N$ "universes." Universes are simply sectors of the theory where the expectation values take the form

$$\langle U_n(x)\rangle_k = \exp\left(ik\frac{2\pi n}{N}\right). \tag{11}$$

Here $\langle\cdot\rangle_k$ denotes the expectation value in any state in the universe with label $k$. Each universe might have multiple (and possibly degenerate) locally-stable Poincaré-invariant vacua.

We will work with Euclidean spacetimes of the form $M = S^1 \times M_S$, where $S^1$ is a thermal circle of circumference $\beta$, and $M_S$ has volume $\mathcal{V}$, so that the spacetime volume is $V = \beta\mathcal{V}$. In general, the universes have distinct vacuum energy densities $\mathcal{E}_k$, as well as distinct spectra of particle excitations (and hence free energies). The partition function of the full theory decomposes into a sum of the partition functions within each universe:

$$Z = \sum_{k=1}^{N} e^{-V\mathcal{F}_k}, \tag{12}$$

where $\mathcal{F}_k = -\frac{1}{V}\log\sum_\ell e^{-\beta E_{k,\ell}}$ is the spacetime free energy density and $E_{k,\ell}$ is the energy of the $\ell$-th energy eigenstate in the $k$-th universe. States belonging to different universes cannot mix even in finite spatial volume, which is why these sectors are called universes rather than superselection sectors.

Before discussing deformations of theories with the LTOs above, we observe that $U_n(x)$ has scaling dimension $\Delta = 0$ with respect to e.g. a short-distance fixed point. To see this, suppose that under a scale transformation $U_n(x) \to U_n'(x) = \lambda^{-\Delta}U_n(\lambda^{-1}x)$. If we assume that

$x$ in Eq. (10) is small enough for the correlator to be well-described by the short-distance fixed point, then a scale transformation maps Eq. (10) to

$$\lambda^{-2\Delta}\langle U_n^\dagger(\lambda^{-1}x)U_n(0)\rangle = \lambda^{-2\Delta}\langle U_n^\dagger(x)U_n(0)\rangle\,,\tag{13}$$

where we have used the topological property of $U_n^\dagger$. Given that scale transformations are a symmetry at the fixed point, the above expression must coincide with Eq. (10), leading to the conclusion that $\Delta = 0$.[2]

The fact that the LTOs have scaling dimension 0 means that any theory with LTOs has relevant deformations. Recall the deformation in Eq. (5), repeated here with a discrete index $n$ for convenience:

$$\Delta_n S = \int d^d x\,\tfrac{1}{2}\left(\Lambda_n^d U_n(x) + \text{h.c.}\right),\tag{5 revisited}$$

where $\Lambda_n$ is a parameter with dimensions of energy. Given a QFT with an original action $S$, the partition function with the deformation can be written as

$$Z = \int \mathcal{D}[\text{fields}]\,e^{-S}\exp\left(-\int d^d x\,\tfrac{1}{2}(\Lambda_n^d U_n(x) + \text{h.c.})\right).\tag{14}$$

Expanding the deformation term gives

$$Z = Z_u \sum_{I=0}^{\infty}\sum_{J=0}^{\infty}\frac{(-\tfrac{1}{2}\Lambda_n^d)^I(-\tfrac{1}{2}(\Lambda_n^d)^\dagger)^J}{I!\,J!}\left[\prod_{i=1}^{I}\prod_{j=1}^{J}\int d^d x_i d^d y_j\right]\left\langle\prod_{i=1}^{I}\prod_{j=1}^{J}U_n(x_i)U_n^\dagger(y_j)\right\rangle_u,\tag{15}$$

where the subscript $u$ indicates the designated quantity is taken to be in the undeformed theory. At this stage, the above expression can be thought of as the partition function of an ideal gas of local topological defects. Note that this does *not* mean that the $(d-1)$-form $\mathbb{Z}_N$ symmetry is being gauged. Gauging a discrete symmetry can be achieved by summing over all insertions of symmetry generators with trivial weights [1], but here the insertions have the non-trivial weights seen in Eq. (15). In particular, if we insert an operator charged under the $(d-1)$-form global symmetry, the sum over defects inside the loop does *not* cause the expectation value to identically vanish.

To proceed, we use the fact that expectation values in the original, undeformed theory can be written as sums of expectation values calculated within the $N$ distinct universes:

$$\left\langle\prod_{i=1}^{I}\prod_{j=1}^{J}U_n(x_i)U_n^\dagger(y_j)\right\rangle_u = \frac{1}{Z_u}\sum_{k=1}^{N}\left\langle\prod_{i=1}^{I}\prod_{j=1}^{J}U_n(x_i)U_n^\dagger(y_j)\right\rangle_{k,u}e^{-V\mathcal{F}_{k,u}}\tag{16}$$

$$= \frac{1}{Z_u}\sum_{k=1}^{N}\langle U_n\rangle_{k,u}^I\langle U_n^\dagger\rangle_{k,u}^J\,e^{-V\mathcal{F}_{k,u}},\tag{17}$$

where $\langle\cdot\rangle_{k,u}$ means an expectation value calculated in the $k$-th universe of the undeformed theory. In Eq. (16) we have used the fact that the expectation value of an LTO is the same in any state within a given universe. The topological property of the $U_n$ operators together with cluster decomposition lead to Eq. (17). Next, using the fact that $\langle U_n\rangle_{k,u} = \exp\left(ik\frac{2\pi n}{N}\right)$, we

---

[2]The fact that LTOs are dimension-0 operators was noted earlier in Ref. [25] in the context of certain CFTs.

land on

$$Z = \sum_{k=1}^{N} e^{-V\mathcal{F}_{k,u}} \sum_{I=0}^{\infty} \frac{(-\frac{1}{2}\Lambda_n^d \exp\left(ik\frac{2\pi n}{N}\right) \int d^d x)^I}{I!} \sum_{J=0}^{\infty} \frac{(-\frac{1}{2}(\Lambda_n^d)^\dagger \exp\left(-ik\frac{2\pi n}{N}\right) \int d^d x)^J}{J!} \tag{18}$$

$$= \sum_{k=1}^{N} e^{-V\mathcal{F}_{k,u}} e^{-|\Lambda_n^d| V \cos(2\pi kn/N + \chi_n)} \tag{19}$$

$$= \sum_{k=1}^{N} \sum_{\ell=0}^{\infty} \exp\left[-\beta\left(E_{k,\ell} + \mathcal{V}|\Lambda_n^d| \cos(2\pi kn/N + \chi_n)\right)\right]. \tag{20}$$

where $\Lambda_n^d = |\Lambda_n^d| e^{i\chi_n}$, So the effect of the deformation is to shift all energies of the states within a given universe by a common amount. But states in different universes get different energy shifts, distinguishing our "universal deformations" from the cosmological constant. For applications to confinement in 2d gauge theory, we note that the vacuum energy densities $\mathcal{E}_k = E_{k,0}/\mathcal{V}$ in each universe become

$$\mathcal{E}_k = \mathcal{E}_{k,u} + |\Lambda_n^d| \cos(2\pi kn/N + \chi_n). \tag{21}$$

The expectation value of a unit-charge test "brane" — that is, a $(d-1)$-dimensional operator charged under the $(d-1)$-form symmetry — in the universe labelled by $k$ is determined by the difference in energy densities inside and outside the loop: $\mathcal{E}_{k+1} - \mathcal{E}_k$.

It is easy to generalize the analysis above to determine the effect of a general LTO deformation

$$\Delta S = \int d^d x\, \frac{1}{2} \sum_{n=1}^{N-1} \left(\Lambda_n^d U_n(x) + \text{h.c.}\right), \tag{22}$$

where we have excluded the $n = 0$ trivial deformation because it shifts all the energies across all the universes by the same amount. This deformation leads to

$$\boxed{\mathcal{E}_k = \mathcal{E}_{k,u} + \sum_{n=1}^{N-1} |\Lambda_n^d| \cos(2\pi kn/N + \chi_n).} \tag{23}$$

As an example, suppose $N = 2$ so that there is only one non-trivial LTO deformation, parameterized by the coefficient $\Lambda_1 \equiv \Lambda$ as in Eq. (22). Suppose that the two universes are not degenerate when $\Lambda = 0$, with vacuum energy densities $\mathcal{E}_{1,u} \neq \mathcal{E}_{2,u}$. This means that the unit-charge test branes are confined at $\Lambda = 0$. But if we dial $\Lambda$ to the special value

$$\Lambda_*^d = \frac{\mathcal{E}_{1,u} - \mathcal{E}_{2,u}}{\cos(2\pi \cdot 2 \cdot 1/2) - \cos(2\pi \cdot 1 \cdot 1/2)} = \frac{\mathcal{E}_{1,u} - \mathcal{E}_{2,u}}{2}, \tag{24}$$

then the vacuum energies in the two universes become degenerate, unit-charge branes become deconfined, and the $(d-1)$-form symmetry is spontaneously broken. This means that in the infinite volume limit we can drive the theory through a first-order phase transition by dialing $\Lambda$.

# 3 Charge-$N$ Schwinger model

Consider the charge-$N$ Schwinger model: a 2d $U(1)$ gauge theory coupled to a charge-$N$ Dirac fermion. This model has been extensively studied recently, see e.g. [9, 28, 30, 33, 39, 43]. The Euclidean action is given by

$$S_\psi = \int d^2x \left( \frac{1}{4e^2} f_{\mu\nu} f^{\mu\nu} + \frac{i\theta}{2\pi} \epsilon^{\mu\nu} \partial_\mu a_\nu + \overline{\psi}(\slashed{D} - m_\psi)\psi \right), \tag{25}$$

where $D_\mu = \partial_\mu - iNa_\mu$ and $f_{\mu\nu} = \partial_\mu a_\nu - \partial_\nu a_\mu$. We will use the convention that $m_\psi \geq 0$. When $m_\psi = 0$, this model has a 0-form $\mathbb{Z}_N$ chiral symmetry which is left unbroken by the axial anomaly, and acts as

$$\psi \to \exp\left( i\gamma_5 \frac{2\pi k}{N} \right)\psi, \tag{26}$$

where $k = 0, \ldots, N-1$. The $\mathbb{Z}_2 \subset \mathbb{Z}_{2N}$ subgroup acting as $\psi \to -\psi$ is fermion parity $(-1)^F$. Since $(-1)^F$ is gauged, the faithfully acting chiral symmetry is $\mathbb{Z}_N = \mathbb{Z}_{2N}/\mathbb{Z}_2$. There is also a 1-form $\mathbb{Z}_N$ symmetry which acts on Wilson loops $W_q(C) = \exp\left( iq \oint_C a \right)$, where $a = a_\mu dx^\mu$. It is generated by local topological operators $U_n(x)$ which have a $\mathbb{Z}_N$ fusion rule and obey Eq. (8). Abstractly, the $U_n(x)$ operators can be thought of as Gukov-Witten operators, defined by inserting a $2\pi n/N$ flux at some point $x$ (that is, $\int_D da = 2\pi n/N$ for an infinitesimal disk $D$ surrounding $x$). We will give a more explicit discussion of the LTOs below.

It is useful to work with the bosonized form of the action, which is given in form notation by

$$S_\varphi = \int_M \left( \frac{1}{2e^2} \|da\|^2 + \frac{1}{8\pi} \|d\varphi\|^2 + \frac{i}{2\pi}(N\varphi + \theta) \wedge da - (\star 1) m\mu \cos\varphi \right), \tag{27}$$

where $\varphi$ is a $2\pi$-periodic scalar dual to $\psi$, $\mu$ is a renormalization scale, $\star 1$ is the volume form, and $\|X\|^2 = X \wedge \star X$ for any form $X$. The parameter $m = \frac{e^\gamma}{2\pi} m_\psi$, where $\gamma$ is the Euler-Mascheroni constant. In the bosonized variables, chiral symmetry acts by shifts $\varphi \to \varphi + 2\pi k/N$ with $k = 0, 1, \ldots, N-1$.

## 3.1 Concrete expressions for LTOs

We now discuss how to write the LTOs of the Schwinger model concretely in terms of fields.[3] In general there is no guarantee that a topological operator should have a simple expression in terms of the fields that appear in some particular path integral representation of a quantum field theory. It turns out that in the variables of $S_\varphi$ (and for that matter $S_\psi$) it is hard to write an explicit representation of the LTOs. We discuss the challenge in Appendix A.

To get a concrete expression for the LTOs, we integrate in a new $\mathbb{R}$-valued scalar field $b$ and switch to the action

$$S_{\varphi,b} = \int_M \left( \frac{e^2}{2} \|b\|^2 + \frac{1}{8\pi} \|d\varphi\|^2 + \frac{i}{2\pi}(N\varphi + 2\pi b + \theta) \wedge da - (\star 1) m\mu \cos\varphi \right). \tag{28}$$

The equation of motion for $b$ is $b = -\frac{i}{e^2} \star da$, so if we integrate out $b$, we recover the original action. Working with $b$, $\varphi$, and $a$ as dynamical field variables lets us write an expression for the $\mathbb{Z}_N$ 1-form symmetry generators as

$$U_n(x) = \exp\left[ i\frac{2\pi n}{N} \left( b + \frac{N}{2\pi}\varphi + \frac{\theta}{2\pi} \right) \right], \qquad n = 0, 1, \ldots, N-1. \tag{29}$$

---

[3]We are grateful to Y. Hidaka for very helpful discussions on how to correctly write topological operators in terms of fields.

Note that $U_n(x)$ has charge $n$ under the $\mathbb{Z}_N$ chiral symmetry when $m = 0$. This is a consequence of the mixed 't Hooft anomaly for the 1-form and 0-form $\mathbb{Z}_N$ symmetries of the model [28]. Similarly, $U_n(x)$ is not invariant under $\theta \to \theta + 2\pi$, reflecting the anomaly in the space of couplings between the $\theta$-periodicity and the 1-form symmetry [44, 45].

We now explain how to verify that Eq. (29) is in fact the correct expression for the local topological operators that generate the $\mathbb{Z}_N$ 1-form symmetry. First, we observe that the equation of motion for $a$ implies that $U_n(x)$ is a constant on-shell. This is a necessary but not sufficient condition for Eq. (29) to define the desired topological operator. We must also check that its expectation value has the expected behavior:

$$\langle U_n(x) \rangle = \frac{1}{Z} \int \mathcal{D}a \, \mathcal{D}b \, \mathcal{D}\varphi \, e^{-S_{\varphi,b}} U_n(x) \tag{30}$$

$$= \frac{1}{Z} \int \mathcal{D}f \, \mathcal{D}b \, \mathcal{D}\varphi \sum_{\nu \in \mathbb{Z}} \delta\left(\nu - \frac{1}{2\pi} \int_M f\right) e^{-S_{\varphi,b}} U_n(x) \tag{31}$$

$$= \frac{1}{Z} \int \mathcal{D}f \, \mathcal{D}b \, \mathcal{D}\varphi \sum_{k \in \mathbb{Z}} \exp\left(ik \int_M f\right) e^{-S_{\varphi,b}} U_n(x) \tag{32}$$

$$= \frac{1}{Z} \int \mathcal{D}f \, \mathcal{D}b \, \mathcal{D}\varphi \sum_{k \in \mathbb{Z}} e^{-S^{(k)}_{\varphi,b,\mathrm{eff}}}, \tag{33}$$

where in the second line we switched variables to $f = da$ and introduced the explicit constraint that the topological charges are quantized, and[4]

$$S^{(k)}_{\varphi,b,\mathrm{eff}} = \int_M \left( \frac{e^2}{2} \|b\|^2 + \frac{1}{8\pi} \|d\varphi\|^2 + \frac{i}{2\pi} (N\varphi + 2\pi b + \theta - 2\pi k) \wedge f \right.$$
$$\left. - \frac{i}{2\pi} (N\varphi + 2\pi b + \theta) \wedge \frac{2\pi n}{N} \delta^{(2)}(x) - (\star 1) m\mu \cos\varphi \right). \tag{34}$$

If we now shift $f \to f + \frac{2\pi n}{N} \delta^{(2)}(x)$, we deduce that

$$\langle U_n(x) \rangle_k = \exp\left(\frac{2\pi ikn}{N}\right). \tag{35}$$

This is indeed how the expectation value of $U_n(x)$ should behave in the $k$-th universe. The same sort of calculation can be used to verify that the $U_n(x)U_m(x) = U_{n+m \bmod N}(x)$ fusion rule is satisfied, as well as to check that $\langle U_{n_1}(x_1)U_{n_2}(x_2)\cdots\rangle$ is independent of the insertion points $x_i$. Finally, we should verify that $U_n(x)$ has the expected behavior in correlation functions with Wilson loops:

$$\left\langle U_n(x) \exp\left(iq \oint_C a\right) \right\rangle = \frac{1}{Z} \int \mathcal{D}a \, \mathcal{D}b \, \mathcal{D}\varphi \left\{ e^{-S_{\varphi,b}} \exp\left(iq \int_M a \wedge \delta^{(1)}(C)\right) \right. \tag{36}$$
$$\left. \times \exp\left(i\frac{2\pi n}{N} \int_M \left[ b + \frac{N}{2\pi} \varphi + \frac{\theta}{2\pi} \right] \wedge \delta^{(2)}(x)\right) \right\}.$$

Changing variables using the substitution $a \to a + \frac{2\pi n}{N} \delta^{(1)}(C_{x,y})$, where $C_{x,y}$ is a smooth simple curve connecting the point $x$ to the point $y$ outside of $C$, we land on:

$$\left\langle U_n(x) \exp\left(iq \oint_C a\right) \right\rangle = \exp\left(\frac{2\pi iqn}{N} \int_M \delta^{(1)}(C_{x,y}) \wedge \delta^{(1)}(C)\right) \left\langle U_n(y) \exp\left(iq \oint_C a\right) \right\rangle$$
$$= \exp\left(\frac{2\pi iqn}{N} \mathcal{I}(C, C_{x,y})\right) \left\langle U_n(y) \exp\left(iq \oint_C a\right) \right\rangle, \tag{37}$$

---

[4]We adopt the convention for delta-function forms that $\int_{M_d} X^{(p)} \wedge \delta^{(d-p)}(\Sigma_p) = \int_{\Sigma_p} X^{(p)}$ for a $p$-form $X$. It follows that $d\delta^{(d-p)}(\Sigma_p) = (-1)^p \delta^{(d-p+1)}(\partial\Sigma_p)$.

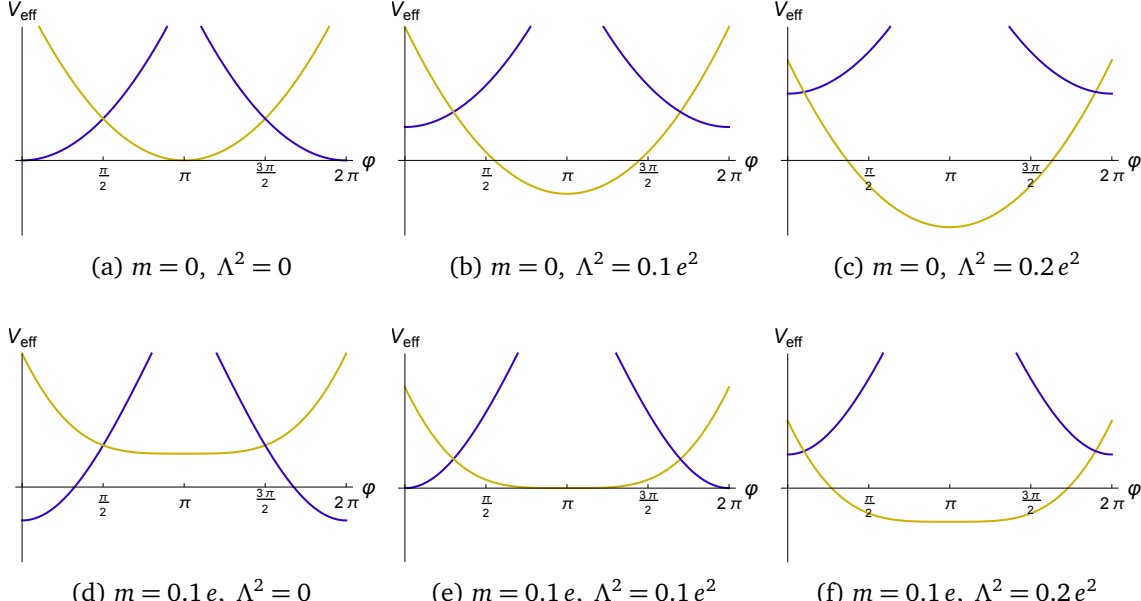

Figure 3: The blue and gold curves in the plots above, with minima at 0 and $\pi$ respectively, are effective potential energy densities in the $k = 0$ and $k = 1$ universes of the charge-2 Schwinger model, displayed as a function of the bosonized scalar $\varphi$ with six different choices of the mass parameter $m$ and universal deformation parameter $\Lambda$. Here we have chosen $\theta = \delta = 0$, and therefore $\chi = 0$. The only effect of the deformation is to shift the relative potential energy density curves in the two universes, without affecting their shape. At $m = 0$, the deformation explicitly breaks chiral symmetry, but no fermion mass term is generated.

where $\mathcal{I}(C, C_{x,y})$ is the intersection number of $C$ and $C_{x,y}$. If $x$ is inside $C$ and $y$ is outside $C$, then the intersection number is just the linking number of $x$ and $C$. This completes our verification that Eq. (29) is a valid representation of the LTOs that generate the 1-form $\mathbb{Z}_N$ symmetry of the charge-$N$ Schwinger model.

## 3.2 Effects of universal deformations

Now consider adding the $U_1(x)$ operator to the Lagrangian,

$$\Delta \mathcal{L} = \tfrac{1}{2} |\Lambda^2| e^{i\chi} e^{i(\varphi + 2\pi b/N + \theta/N)} + \text{h.c.}, \tag{38}$$

where $\chi$ is an arbitrary phase. For a fixed $\chi$, the resulting path integral is only periodic under $\theta \to \theta + 2\pi N$. This means that the 2d charge-$N$ Schwinger model with $m > 0$ and an LTO deformation with a $\theta$-independent phase happens to have the same $\theta$ periodicity as the charge-$N$ Schwinger model with a gauged $\mathbb{Z}_N$ 1-form symmetry.[5] Of course, one could instead choose to set $\chi = \delta - \theta/N$, and then the path integral is invariant under $\theta \to \theta + 2\pi$ and $\delta \to \delta + 2\pi$. The theory is invariant under charge conjugation (which flips the sign of $a, \varphi, b$) if $\theta = 0, \pi$ and $\delta = 0, \pi$.

To keep the discussion simple, let us take $N = 2$, so that there is only one non-trivial deformation $U_1(x)$. We can understand the impact of the deformation by appealing to the general calculation in Sec. 2. Alternatively, we can add the explicit form of $U_1(x)$ given in Eq. (29) to the action. Integrating out $a$ while being careful to ensure that $\int_M f \in 2\pi\mathbb{Z}$ gives a

---

[5]Despite the fact that the deformation breaks chiral symmetry, the form of the one-form symmetry generator is such that when $m = 0$, $\theta$ is still unphysical and observables only depend on $\chi$.

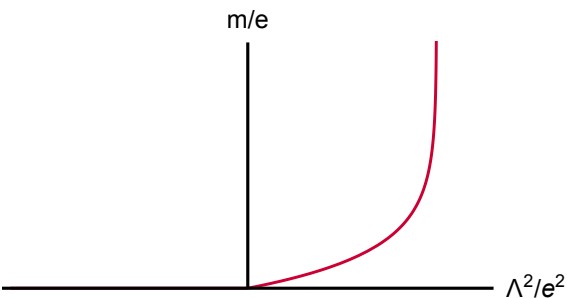

Figure 4: A sketch of the phase diagram of the charge-2 Schwinger model as a function of the fermion mass term and the local topological operator deformation, with coefficients $m$ and $\Lambda^2$ respectively. Here we have chosen $\theta = \delta = 0$, and therefore $\chi = 0$. The theory is in a confined phase in the bulk of the plot, but it is deconfined (with spontaneously broken $\mathbb{Z}_2$ 1-form symmetry) along the red curve.

functional delta function which allows us to eliminate $b$ from the effective action. Of course, both approaches lead to the same result. The effective potential energy at the scale $\mu = e$ and with $\theta = \delta = 0$ is

$$V_k(\varphi) = \frac{1}{2}\left(\frac{eN}{2\pi}\right)^2\left(\varphi - \frac{2\pi k}{N}\right)^2 - me\cos\varphi + \Lambda^2\cos\left(\frac{2\pi k}{N}\right). \tag{39}$$

The parameter $k$ mod 2 is the universe label. We show plots of $V_0(\varphi)$, $V_1(\varphi)$, and $V_2(\varphi)$ with $m = 0, 0.1\,e$ and $\Lambda^2 = 0, 0.1\,e^2, 0.2\,e^2$ in Fig. 3. These plots show that dialing the coefficient of the universal deformation allows us to change which universe has the lower vacuum energy. This means that as we dial (in charge-conjugation-invariant way) two of the non-trivial continuous parameters of the deformed $N = 2$ Schwinger model, namely $m$ and $\Lambda^2$, we encounter a phase boundary. We sketch the phase diagram as a function of $m > 0$ and $\Lambda^2$ in Fig. 4. Note that if we set $\theta = \pi$ and $\delta = 0, \pi$, the two universes are degenerate and related by charge-conjugation for any $|m|$ and $|\Lambda^2|$. This is a consequence of the mixed anomaly between charge-conjugation and the $\mathbb{Z}_2$ 1-form symmetry at $\theta = \pi$ [27].

The $m = 0$ plots in Fig. 3 illustrate that the $U_1(x)$ deformation breaks chiral symmetry, as expected from the form of Eq. (38). This is a consequence of the 't Hooft anomaly between the $\mathbb{Z}_2$ 1-form and 0-form symmetries. Chiral symmetry acts by $\varphi \to \varphi + \pi$ at $N = 2$, and requires that the $k = 0$ and $k = 1$ universes be degenerate in the undeformed, massless theory. The deformation by $U_1(x)$ clearly breaks the degeneracy of the chiral vacua.

The $U_1(x)$ deformation in the Schwinger model breaks chiral symmetry in precisely the same way as a mass term. If we start with $m = 0$ and turn on such a deformation with coefficient $\Lambda^2$, the principle of effective field theory naturalness suggests that a fermion mass term ought to be generated radiatively. One would expect that $m$ gets driven to the non-zero value $m \sim \Lambda$ after fluctuations are taken into account. However, a very interesting feature of the $U_1(x)$ deformation is that this does *not* happen. This follows because the deformation does not affect the spectrum of excitations — it only affects the relative vacuum energies of the universes. On the other hand, a fermion mass term would affect both the spectrum and the relative vacuum energies. The fact that the deformation does not affect the spectrum of excitations implies that it does not generate a fermion mass term. This is an unusual and interesting violation of the EFT naturalness principle. For a further discussion of this observation see Sec. 5.

# 4 Extension to non-invertible local topological operators

So far we have considered deformations by LTOs associated with invertible $(d-1)$-form symmetries. Extending this to non-invertible symmetries is straightforward. At first glance, there is a crucial difference between invertible and non-invertible LTOs — the former satisfy $\langle U_a(x) U_b(y)\rangle = \langle U_a(x)\rangle\langle U_b(y)\rangle$ for *any* $x$ and $y$, while such factorization only holds for non-invertible operators when $x \neq y$. Indeed, when evaluated at coincident points, generic non-invertible LTOs satisfy a non-trivial fusion rule, see Eq. (4). Since we relied on factorization of LTO expectation values in order to evaluate the effects of the deformations explicitly, one might wonder whether such an analysis breaks down for deformations by non-invertible LTOs.

This turns out not to be an issue in the continuum limit. At each given order in the expansion of the LTO deformation, contributions involving $\ell$ operators at coincident points appear with an additional factor $\sim (V \Lambda_{\mathrm{UV}}^d)^{1-\ell}$ relative to contributions that do not involve coincident points. The UV scale $\Lambda_{\mathrm{UV}}$ can be thought of as the inverse lattice spacing $1/a$ — in a dilute-gas picture, $\Lambda_{\mathrm{UV}}$ can be thought of as an "inverse size" of the LTOs. After re-exponentiation, the extra shift in the free energy density coming from the contributions of $m$ coincident operators scales as $\Delta \mathcal{F}_k \sim \Lambda^d (\Lambda/\Lambda_{\mathrm{UV}})^{d(\ell-1)}$. This implies that it is safe to neglect the contributions of coincident operators provided the deformation scale $\Lambda$ is well separated from the UV cutoff.

As a concrete example, let us consider pure $SU(N)$ Yang-Mills theory in two spacetime dimensions. The partition function of the theory defined on a surface with area $A$ and genus $g$ is given by [46, 47]

$$Z = \sum_r d_r^{2-2g}\, e^{-c_r e^2 A}, \tag{40}$$

where $e$ is the gauge coupling, $r$ is a unitary irreducible representation of $SU(N)$, and $d_r, c_r$ are the dimension and quadratic Casimir of $r$, respectively. The above partition function takes the form of a sum over universes (one for each irrep), where each universe consists of a single vacuum state with no additional excitations, as gluons have no propagating degrees of freedom in two spacetime dimensions. As one might expect given the existence of universes, this theory has LTOs [18]. They can be thought of as Gukov-Witten operators [48–50], which are defined by constraining the conjugacy class of the holonomy of the gauge field around insertion points in spacetime. The Gukov-Witten operators are labelled by conjugacy classes $\omega$ of $SU(N)$, and their expectation values in the universe $r$ are simply

$$\langle U_\omega\rangle_r = \frac{\chi_r(\omega)}{d_r}, \tag{41}$$

where $\chi_r(\omega)$ is the character in representation $r$. In infinite spacetime volume the operators $U_\omega$ satisfy the following relation with Wilson loops,

$$U_\omega(x) W_r(C) = \frac{\chi_r(\omega)}{d_r} U_\omega(\tilde{x}) W_r(C), \tag{42}$$

where the Wilson loop trace is taken in the irreducible representation $r$, and we have assumed that $x$ links the contour $C$ while $\tilde{x}$ does not. Two Gukov-Witten operators at the same spacetime point fuse according to

$$U_\omega(x) U_\delta(x) = \sum_\rho N_{\omega\delta}^\rho\, U_\rho(x), \tag{43}$$

where $N_{\omega\delta}^\rho$ is roughly the number of ways of multiplying an element in $\omega$ with an element in $\delta$ to obtain an element in $\rho$ [15].[6] Generic Gukov-Witten operators are non-invertible. If $\omega$ is

---

[6] In $SU(N)$, the conjugacy classes are labeled by continuous parameters specifying a location on the maximal torus, so the sum in Eq. (43) is really an integral. It would be interesting to work out the fusion coefficients in $SU(N)$ explicitly.

the conjugacy class of an element in the center of $SU(N)$, then $U_\omega$ is an invertible symmetry operator for the $\mathbb{Z}_N$ 1-form symmetry of pure YM theory.

As in the invertible case, we now consider deforming the theory by a (potentially non-invertible) LTO $U_\omega$,

$$\Delta_\omega S = \int_{M_2} d^2x \, \tfrac{1}{2} \Lambda_\omega^d \left( U_\omega(x) + U_\omega^\dagger(x) \right), \tag{44}$$

where $U_\omega^\dagger = U_{\omega^\dagger}$ is the Gukov-Witten operator corresponding to the conjugacy class of Hermitian conjugates of elements in $\omega$ (we take $\Lambda_\omega^d$ to be real for simplicity). In order to systematically treat contributions to the deformed partition function from coincident LTOs, we regularize the theory on a lattice with spacing $a$ and the heat-kernel action [18, 46, 47]. This action has a remarkable "subdivision" property which implies that the results one obtains at finite lattice spacing automatically coincide with the results in the continuum limit of 2d $SU(N)$ YM theory. The partition function of the deformed theory is

$$Z = \int \prod_\ell du_\ell \prod_p \sum_r d_r \chi_r(u_p) e^{-c_r e^2 a^2} \prod_x e^{-\frac{1}{2}\Lambda_\omega^2 a^2 \left( U_\omega(x) + U_{\omega^\dagger}(x) \right)}, \tag{45}$$

where $r$ runs over irreducible representations, $\ell$ runs over the links of the lattice, $u_\ell$ are the $SU(N)$-valued link variables, $u_p$ is the path-ordered product of link variables around the boundary of the plaquette $p$, and $x$ are sites on the dual lattice. We denote by $\mathcal{A}$ the number of plaquettes on the lattice. Following the steps in Sec. 2 but omitting contributions from coincident points, we find

$$Z \approx Z_u \sum_{I+J \leq \mathcal{A}} \frac{(-\frac{1}{2}\Lambda_\omega^2 a^2)^{I+J}}{I!J!} \sum_{\substack{x_1,\ldots,x_I, y_1,\ldots,y_J \\ x_i \neq x_j \neq y_k \neq y_\ell}} \left\langle \prod_{i=1}^I \prod_{j=1}^J U_\omega(x_i) U_{\omega^\dagger}(y_j) \right\rangle_u$$

$$= \sum_r d_r^{2-2g} \, e^{-c_r e^2 a^2 \mathcal{A}} \sum_{I+J \leq \mathcal{A}} \binom{\mathcal{A}}{I} \left( -\tfrac{1}{2}\Lambda_\omega^2 a^2 \frac{\chi_r(\omega)}{d_r} \right)^I \binom{\mathcal{A}-I}{J} \left( -\tfrac{1}{2}\Lambda_\omega^2 a^2 \frac{\chi_r(\omega^\dagger)}{d_r} \right)^J. \tag{46}$$

In the large area limit we can perform the above sums independently and we find, as expected, that the vacuum energy densities in each universe get shifted,

$$\mathcal{E}_r = e^2 c_r + \tfrac{1}{2}\Lambda_\omega^2 \left( \frac{\chi_r(\omega) + \chi_r(\omega^\dagger)}{d_r} \right). \tag{47}$$

The lattice regularization also allows us to explicitly compute the contributions from coincident operators. Applying the procedure above to include pairs of operators at coincident points, we find an additional shift in the vacuum energies

$$\Delta \mathcal{E}_r = \tfrac{1}{4}\Lambda_\omega^2 (\Lambda_\omega a)^2 \sum_\rho \left( N_{\omega\omega}^\rho + 2N_{\omega\omega^\dagger}^\rho + N_{\omega^\dagger\omega^\dagger}^\rho \right) \frac{\chi_r(\rho)}{d_r}. \tag{48}$$

As expected from the general discussion above, this contribution is suppressed relative to (47) in the continuum limit where we take $a \to 0$ and $\mathcal{A} \to \infty$ with $\Lambda_\omega$ and $a^2\mathcal{A}$ held fixed. In the defect gas picture, this suppression corresponds to the fact that the non-invertible LTOs can be viewed as having contact interactions but effectively zero size, so that the gas becomes extremely dilute and weakly interacting as we take the continuum limit of the deformed theory.

Equation (47) suggests that it might be possible to choose a collection of deformations to make any given set of Wilson loops deconfined in 2d YM theory. For example, if we want to make the fundamental and adjoint Wilson loops deconfined in 2d $SU(2)$ YM, we can add LTO

deformations associated to the conjugacy classes of $-\mathbf{1}$ and $i\sigma_3$ with coefficients $-3e^2/8$ and $3e^2/2$ respectively.[7] The $i\sigma_3$ deformation is associated with a non-invertible LTO. With these two deformations, the fundamental-representation (spin $s = 1/2$) and adjoint-representation ($s = 1$) universes become degenerate with the trivial-rep universe. Similarly, if we want the $s = 1/2, 1, 3/2$ representation Wilson loops to be deconfined, this can be done with deformations by LTOs associated with $e^{i\theta\sigma_3}$ with $\theta = \pi, \pi/2, \pi/3$ with coefficients $e^2/8, -3e^2/2, 4e^2$. We have checked numerically that it is possible to make all test-charge representations up to spin $s = s_{\max}$ deconfined up to $s_{\max} = 4$, with no apparent obstruction to pushing $s_{\max}$ up as much as one likes, at the cost of introducing $2s_{\max}$ distinct deformations.

## 5 Symmetry breaking and naturalness

We have seen that QFTs with local topological operators (LTOs) necessarily have exactly-solvable relevant deformations, which are obtained by simply adding the LTOs to the action. The effect of these "universal deformations" is to allow one to continuously dial the vacuum energy densities of the universes associated with the existence of the LTOs. The LTOs generate a $(d-1)$-form symmetry, and the comparative vacuum energy densities of universes determine its realization. This means that the realization of $(d-1)$-form symmetries is sensitive to the deformations we have introduced, and by dialing the strength of the deformations we can drive a QFT through phase transitions.

One interesting immediate implication of our results involves the Coleman-Mermin-Wagner theorem [51,52]. This theorem constrains the realization of conventional 0-form global symmetries, so that e.g. discrete 0-form symmetries cannot be spontaneously broken in spacetime dimension $d < 2$. In their work introducing $n$-form global symmetries [1], Gaiotto et al. gave an argument that the Coleman-Mermin-Wagner theorem can be extended to the statement that discrete $n$-form symmetries cannot be spontaneously broken when $d - n < 2$, see also Ref. [53].

There are two ways to probe whether an $n$-form symmetry is spontaneously broken. One approach involves studying the behavior of charged operators $W(M_n)$ where $M_n$ is topologically trivial but large, while the other approach involves studying $W(M_n)$ where $M_n$ is a homologically non-trivial $n$-cycle of the spacetime manifold.[8] For example, for 1-form symmetries the first approach is to ask whether the expectation values of charged operators $W(C)$ have an area-law or a perimeter-law, and to interpret the latter as a sign of spontaneous symmetry breaking [1]. With this perspective, there is a direct connection between the realization of the 1-form symmetry and the question of whether some charged probe particles are confined or deconfined in Minkowski space, in infinite spatial volume. The second approach, which goes back to the 1980s literature on center symmetry [54,55], is to rotate to Euclidean space, compactify the time direction to a circle $S_\beta^1$, and ask about the expectation value of the Polyakov loop $W(S_\beta^1)$. Then one says that the 1-form "center" symmetry is spontaneously broken if $\langle W(S_\beta^1) \rangle \neq 0$. Reference [1] used this second perspective in discussing the generalization of the Coleman-Mermin-Wagner theorem for discrete $n$-form symmetries.

There is good evidence that the two approaches discussed above are equivalent for exploring the realization of discrete $n$-form symmetries provided $0 < n < d - 1$. But later developments showed that they are not equivalent for $(d-1)$-form symmetries. As a result, there are

---

[7]Here by deconfined we mean that the expectation value of the Wilson loop in a given representation has perimeter-law in the universe corresponding to the trivial representation. In the examples we consider, the trivial universe is always (one of) the universe(s) with lowest energy density.

[8]We are grateful to Z. Komargodski for very helpful comments on these issues, as well as to M. Shifman for helpful discussions on the Schwinger model on a cylinder.

some counterexamples to an extension of the Coleman-Mermin-Wagner theorem to discrete $n$-form symmetries. For instance, consider the charge-2 Schwinger model or 2d $SU(2)$ YM coupled to a massless adjoint Majorana fermion. Each of these $d = 2$ QFTs has a $\mathbb{Z}_2$ 1-form symmetry which participates in a mixed 't Hooft anomaly with a $\mathbb{Z}_2$ 0-form chiral symmetry, see e.g. [28, 31, 56], so the expectation value of the fundamental Wilson line has a perimeter-law fall-off. This means that test particles in the fundamental representation are deconfined, and the 1-form symmetry is spontaneously broken. At the same time, if we compactify space to a circle with periodic boundary conditions, the effective quantum mechanical description features exactly-degenerate ground states thanks to the 't Hooft anomaly mentioned above. The expectation value of the Polyakov loop could be zero or non-zero depending on the particular linear combination of ground states in which it is evaluated. This illustrates that the traditional Polyakov loop criterion for confinement is potentially ambiguous for $(d - 1)$-form symmetries.

The discussion in the paragraph above might make it tempting to conjecture that a generalization of the Coleman-Mermin-Wagner theorem should hold provided the discrete symmetries in question are not involved in 't Hooft anomalies. However, the results in this paper imply that this is also not a viable candidate for a theorem, because the realization of $(d - 1)$-form symmetries is also sensitive to universal deformations. Nevertheless, in all of the examples of 2d QFTs with 1-form symmetries that we are aware of, the 1-form symmetry is not spontaneously broken in the absence of universal deformations and relevant mixed 't Hooft anomalies. So perhaps there is some appropriately refined version of a Coleman-Mermin-Wagner theorem for $n$-form symmetries after all.

Another interesting implication of our work concerns the EFT naturalness principle. The EFT naturalness principle states that all operators that are not forbidden by a symmetry of the low-energy theory will be generated in the low-energy effective action by quantum fluctuations, with a scale determined by the scale of the operators that break the symmetry. Much of modern particle physics phenomenology is dedicated to exploiting the EFT naturalness principle or looking for ways around it. Our universal deformations lead to an apparently novel violation of this principle when a $(d - 1)$-form symmetry has a mixed 't Hooft anomaly with a 0-form symmetry.

Let us again consider the charge-2 Schwinger model or 2d $SU(2)$ YM coupled to an adjoint Majorana fermion (2d adjoint QCD). In the massless limit, these models each have an LTO $U_1(x)$ that generates a $\mathbb{Z}_2$ 1-form symmetry, as well as an 't Hooft anomaly which forces $U_1(x)$ to carry unit charge under a 0-form $\mathbb{Z}_2$ chiral symmetry. So if we deform the action by $\frac{1}{2}\Lambda^2 \int d^2x \, (U_1(x) + \text{h.c.})$, chiral symmetry is explicitly and completely broken at the scale $\Lambda$. The naturalness principle would then predict that a fermion mass term $\sim m_\psi \int d^2x \, \overline{\psi}\psi$ with $m_\psi \sim \Lambda$ should be generated by quantum fluctuations. However, our analysis above has shown that the *only* effect of the LTO deformation is on the relative vacuum energies of universes. The particle spectra are not affected, and thus a fermion mass term is not generated.

It is also interesting to turn things around and turn on a fermion mass but not the LTO deformations in the UV action. Then LTO operators are radiatively generated in the long-distance effective action. To see this, note that a standard QFT "calculation" of the vacuum energy amounts to summing up the energies of all excitations, and this happens independently within each universe. Since the spectrum of excitations in different universes becomes non-degenerate when the fermion mass is non-zero, one needs to turn on the LTO deformation operators as counter-terms, and then their finite pieces show up in the low-energy effective action. While on this level the naturalness principle seems to work in this direction, there is still an imprint of the peculiar behavior highlighted in the preceding paragraph. When one turns on a fermion mass as well as the deformation by $U_1(x)$ one would expect the fermion mass to be additively renormalized, but this does not happen.

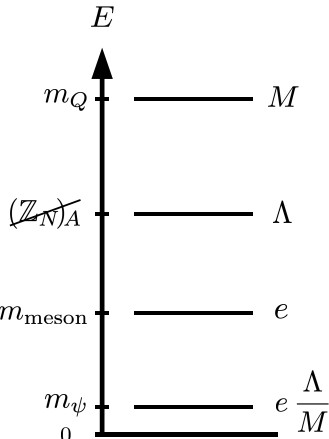

Figure 5: A sketch of some energy scales in 2d $U(1)$ gauge theory with gauge coupling $e$ with a charge $N$ fermion $\psi$ with zero mass in the bare action. We also add a charge 1 field $Q$ with a large mass $M \gg e$. The scale $e$ can be thought of as a kind of "meson" ($\overline{\psi}$-$\psi$ bound state) mass scale. Imagine we add a relevant deformation by an approximately-topological local operator with a coefficient $\Lambda$, with the hierarchy $e \ll \Lambda \ll M$, which is charged under the chiral symmetry $(\mathbb{Z}_N)_A$ of the charge $N$ fermion. Normally one would expect that this would induce a mass term of size $\Lambda$. But for reasons discussed in the text, the mass term has to vanish if $M \to \infty$, so $m_\psi$ has to be proportional to a positive power of $\Lambda/M$. In the sketch we show the most naive possibility $m_\psi \sim e\Lambda/M$.

All this leads us to conclude that EFT naturalness can apparently fail in QFTs with $(d-1)$-form symmetries. It would be satisfying to understand this more deeply. Perhaps there is some currently unknown unconventional symmetry which would rescue the naturalness principle. For example, Ref. [9] recently found another apparent violation of naturalness in 2d adjoint QCD and showed that it was due to non-invertible 0-form symmetries. In 2d, $SU(N)$ adjoint QCD with $N > 2$ has two classically marginal four-fermion operators that are allowed by all conventional symmetries [31]. But the non-invertible symmetries discovered in Ref. [9] forbid one of these four-fermion operators, explaining why it is not generated by fluctuations if its coefficient is set to zero in the UV Lagrangian. If the mass term in e.g. the charge-$N$ Schwinger model is charged under some currently unappreciated non-invertible 0-form symmetry, while the LTOs are invariant, the apparent violation of naturalness we discussed here would have an explanation in the same spirit as in Ref. [9]. For now, of course, this scenario is pure speculation.

In this paper we have focused on "universal" relevant deformations of QFTs with exact $(d-1)$-form symmetries. But what is the effect of these deformations if a $(d-1)$-form symmetry is only approximate, due to the presence of charged matter fields (or branes) with a large mass (or tension) $M$? Breaking the symmetry explicitly (even mildly) connects the various universes, and one should expect the formerly-universal deformations to affect both the spectra of excitations and the vacuum energy densities. These effects will no longer be exactly calculable. But if the large $M$ limit is smooth, the dominant effect of the deformations should be on the energy densities of the vacua of the models, with decreasing effects on the spectra of excitations in each vacuum as $M$ is increased. This suggests that the effects we have discussed can lead to the appearance of anomalously low mass scales. Consider e.g. a 2d theory with a natural energy scale $e$ and an emergent 1-form symmetry below some high scale $M \gg e$. Suppose this emergent 1-form symmetry has a mixed 't Hooft anomaly with a chiral symmetry that acts on a massless fermion $\psi$. If we turn on a deformation like Eq. (5) with $e \ll \Lambda \ll M$, the

chiral symmetry would be broken at the scale $\Lambda$. The expectation based on naturalness is that the field $\psi$ should pick up a mass $m_\psi \sim \Lambda$. But our discussion above suggests that instead we should find that $m_\psi$ is suppressed by $M$, so that for example one could find $m_\psi \sim e\Lambda/M \ll e$. This idea is illustrated in Fig. 5.

We should emphasize that the failure of naturalness and the appearance of anomalously small scales described above is not limited to QFTs in two spacetime dimensions. There are examples of QFTs in four spacetime dimensions with $\mathbb{Z}_p$ 3-form and $\mathbb{Z}_p$ 0-form global symmetries with 't Hooft anomalies, see e.g. [32,39]. Our comments on naturalness and anomalously small mass scales apply to these models as well. We hope to report on a quantitative investigation of these effects in two and four spacetime dimensions in forthcoming work.

Finally, we should note some further implications of our results for 2d adjoint QCD, see Refs. [9,31,56–90] for studies of its properties. When $N > 2$ and the fermion mass $m_\psi = 0$, the model has three $\mathbb{Z}_2$ 0-form symmetries (fermion parity, chiral parity, and charge conjugation) and a $\mathbb{Z}_N$ 1-form "center" symmetry. These symmetries are involved in a somewhat rich set of 't Hooft anomalies [31]. With the assumptions that these are the only symmetries of the model, the anomaly-matching arguments in Ref. [31] imply that the theory ought to feature deconfinement of test charges in representations with $N$-ality $N/2$ when $N$ is even, and confinement for all other test charges with non-vanishing $N$-ality. However, Ref. [9] pointed out that there can be further non-invertible 0-form symmetries at the $m_\psi = 0$ point. The topological lines generating these more exotic symmetries are charged under the $\mathbb{Z}_N$ 1-form symmetry. This can be interpreted as an 't Hooft anomaly between the non-invertible symmetry and the 1-form symmetry. If one sits at the locus in parameter space where the non-invertible lines are indeed topological, then adjoint QCD does not confine test charges in any representation.

But what does it take to tune the model to the locus with the extra exotic symmetries? Reference [31] emphasized that the parameter space of 2d adjoint QCD includes the coefficients of two (marginally) relevant four-fermion operators which are consistent with all of its conventional symmetries. Reference [9] observed that one has to tune the coefficient of one of these four-fermion operators to zero for the non-invertible lines to have a chance of being topological. Indeed, for odd $N$ the four-fermion operators are neutral under the conventional global symmetries [31], so one has to rely on the non-invertible symmetries to prevent them from being radiatively generated. Our results here show that actually one has to do much more fine-tuning to land on the version of the theory with the non-invertible topological line operators. If one sits at a point in parameter space where the universes related by the 1-form symmetry are degenerate and subsequently adds local topological operators to the action, generically the universes become non-degenerate. This means that the non-invertible lines do not commute with the LTOs, and one has to tune the coefficients of LTOs in the action to zero in order for the non-invertible lines to be topological.[9] It is also interesting to note that the existence of LTO deformations implies that in general one cannot directly connect the spectrum of particle excitations to the behavior of large Wilson loops in 2d gauge theories. If one *defines* confinement by an area-law behavior for expectation values of large Wilson loops, as the modern perspective on 1-form symmetries suggests one should, apparently one cannot decide whether or not a 2d QFT is confining by studying its particle excitations.

---

[9]Any lattice simulations of 2d adjoint QCD that aims to test whether the theory deconfines would have to deal with this issue.

## Acknowledgments

We are grateful to Tony Gherghetta, Simeon Hellerman, Yoshimasa Hidaka, Zohar Komargodski, Mendel Nguyen, Srimoyee Sen, Eric Sharpe, Semyon Valgushev, and Mithat Ünsal for helpful discussions and comments. We thank the University of Minnesota for support.

## A   Explicit expressions for topological operators

Here we describe a concrete way to write down topological symmetry operators in terms of fields.[10] As an illustrative example, consider the $2\pi$-periodic compact scalar in $d = 2$ spacetime dimensions with action

$$S = \int_{M_2} \frac{1}{2} \|d\varphi\|^2. \tag{49}$$

The quantum field theory defined by a path integral based on Eq. (49) is known to have two 0-form $U(1)$ symmetries associated with the conserved currents

$$j^{(1)} = i\, d\varphi\,, \qquad \tilde{j}^{(1)} = -\frac{1}{2\pi} \star d\varphi\,. \tag{50}$$

This means that it should have two collections of topological line operators, $U_\alpha$ and $\tilde{U}_\beta$, each of them obeying a $U(1)$ fusion rule of the form $U_\alpha(C)U_{\alpha'}(C) = U_{\alpha+\alpha'}(C)$. The operator $U_\alpha(C)$ implements the shift symmetry of $\varphi$, and acts by multiplying charge-$n$ local operators $e^{in\varphi}$ by $e^{in\alpha}$ when $C$ winds once around the charged operator. The operator $\tilde{U}_\beta(C)$ acts on vortex operators.

Let us focus on the $U(1)$ symmetry which acts on $e^{in\varphi}$. It is tempting (and indeed, customary) to define the topological line operator generating the $U(1)$ 0-form shift symmetry by

$$U_\alpha(C) \stackrel{?}{=} \exp\left(i\alpha \oint_C \star j^{(1)}\right), \tag{51}$$

where $C$ is a closed loop in spacetime, and $j^{(1)}$ is defined by Eq. (50). Since $d \star j^{(1)} = 0$ one expects that the above operator depends only topologically on $C$. To conclude that Eq. (51) is a valid representation of the topological line operators that generate the $U(1)$ 0-form shift symmetry, it is important to check that it satisfies the expected fusion rules in correlation functions. To explore this, let us compute $\langle U_\alpha(C)\rangle$ using the definition in Eq. (51):

$$
\begin{aligned}
\left\langle \exp\left(i\alpha \oint_C \star j^{(1)}\right)\right\rangle &= \frac{1}{Z}\int \mathcal{D}\varphi \, \exp\left(-\int_{M_2} \frac{1}{2}\|d\varphi\|^2 + \alpha \star d\varphi \wedge \delta^{(1)}(C)\right) \\
&= \exp\left(\frac{\alpha^2}{2}\int_{M_2} \delta^{(1)}(C)\wedge\star\delta^{(1)}(C)\right)\frac{1}{Z}\int \mathcal{D}\varphi' \, \exp\left(-\int_{M_2}\frac{1}{2}\left\|d\varphi'\right\|^2\right) \\
&= \exp\left(\frac{\alpha^2}{2}\int_{M_2}\delta^{(1)}(C)\wedge\star\delta^{(1)}(C)\right).
\end{aligned}
\tag{52}
$$

Following Ref. [38], to get to the second line we performed the field redefinition $\varphi = \varphi' + \alpha\,\delta^{(0)}(D)$ where $D$ is a surface with boundary $C$, so that $d\varphi = d\varphi' + \alpha\,\delta^{(1)}(C)$. This expectation value has a UV divergence localized on the curve $C$. This amounts to a perimeter-law behavior for the expectation value, so Eq. (51) is not a topological line operator in the quantum field theory.

---

[10]We are grateful to Y. Hidaka for illuminating discussions about this approach.

It may be tempting to try to fix this issue by switching to renormalized operators with unit expectation values,

$$U_{\alpha,r}(C) = \exp\left(-\frac{\alpha^2}{2}\int_{M_2}\delta^{(1)}(C)\wedge\star\delta^{(1)}(C)\right)\exp\left(i\alpha\oint_C\star j^{(1)}\right). \tag{53}$$

These renormalized symmetry operators have the expected action on charged operators,

$$\langle U_{\alpha,r}(C)\,e^{iq\varphi}\rangle = e^{iq\alpha\,\ell(C,x)}\langle e^{iq\varphi}\rangle. \tag{54}$$

While this might look encouraging, there is a problem: the renormalized operators do not satisfy the expected group composition law, and they do not have completely topological correlation functions. Instead, we find

$$\langle U_{\alpha,r}(C)U_{\beta,r}(C)\rangle = \exp\left(\alpha\beta\int_{M_2}\delta^{(1)}(C)\wedge\star\delta^{(1)}(C)\right)\langle U_{\alpha+\beta,r}(C)\rangle. \tag{55}$$

In other words, removing the divergence in Eq. (52) by a rescaling does not give rise to topological operators with the expected fusion properties. Indeed, Eq. (55) implies that the operators in Eq. (53) are still not quite topological. One perspective on this is to give up on the fusion rule as presented in Eq. (2), and instead define the fusion of $U_{\alpha,r}(C)$ and $U_{\beta,r}(C)$ as the result of bringing $U_{\alpha,r}(C)$ and $U_{\alpha,r}(C')$, with $C\cap C'=0$, close together without actually allowing $C$ and $C'$ to coincide.[11] This is the way operator products of non-topological operators are usually defined in QFT, and would work for many purposes. But this approach would cause serious technical complications if we tried to use it in the main text, for reasons explained below Eq. (59).

To avoid these issues and write down a valid representation of the $U(1)$ topological line operators, one needs a different representation of the field theory. The issue above arose because the action was quadratic in $d\varphi$. We can define an equivalent theory with an action which is linear in $d\varphi$ by integrating in an $\mathbb{R}$-valued 1-form field $b^{(1)}$ and taking the action to be

$$S = \int_{M_2}\left[\frac{1}{2}\left\|b^{(1)}\right\|^2 + ib^{(1)}\wedge d\varphi\right]. \tag{56}$$

One can think of $b^{(1)}$ as a momentum associated to $\varphi$, and if we eliminate $b^{(1)}$ via its local equation of motion $b^{(1)} = i\star d\varphi$ we would get back to Eq. (49). But if we stick to using Eq. (56) as the action, and notice that in these variables the current for the shift symmetry of $\varphi$ is $\star j^{(1)} = b^{(1)}$, we can define the desired topological line operator by

$$U_\alpha(C) = \exp\left(i\alpha\oint_C b^{(1)}\right). \tag{57}$$

With this definition, the expectation value of $U_\alpha(C)$ is finite and independent of $\alpha$:

$$\left\langle\exp\left(i\alpha\oint_C b^{(1)}\right)\right\rangle = \frac{1}{Z}\int\mathcal{D}\varphi\,\mathcal{D}b^{(1)}\exp\left(-\int_{M_2}\frac{1}{2}\left\|b^{(1)}\right\|^2 + ib^{(1)}\wedge\left(d\varphi-\alpha\,\delta^{(1)}(C)\right)\right)$$

$$= \frac{1}{Z}\int\mathcal{D}\varphi'\,\mathcal{D}b^{(1)}\exp\left(-\int_{M_2}\frac{1}{2}\left\|b^{(1)}\right\|^2 + ib^{(1)}\wedge d\varphi'\right) = 1, \tag{58}$$

where again we have shifted $\varphi = \varphi' + \alpha\,\delta^{(0)}(D)$ with $\partial D = C$. One can also verify that $U_\alpha U_\beta = U_{\alpha+\beta}$ and $\langle U_\alpha(C)e^{iq\varphi(x)}\rangle = \exp(iq\alpha\,\ell(C,x))\langle e^{iq\varphi}\rangle$, where $\ell(C,x)$ is the linking number of $C$ and $x$.

---

[11]We thank M. Nguyen and M. Ünsal for discussions on this issue.

This construction generalizes in numerous ways. First, we can consider Lagrangians (in any dimension $d$) which are generic reasonable functions $F(\|d\varphi\|^2)$ plus terms linear in $d\varphi$. One simply writes the "kinetic" term as $F(\left\| b^{(1)} \right\|^2)$ along with a dynamical $(d-1)$-form Lagrange multiplier field which constrains $b^{(1)} = i \star d\varphi$.

Second, it is easy to generalize these observations to $n$-form symmetries. In Sec. 3 of the main text we explained the construction of topological operators for a 1-form symmetry in the 2d Schwinger model. If we had attempted to use the analog of Eq. (51), namely

$$U_n(x) \stackrel{?}{=} \exp\left( i\frac{2\pi n}{N}\left[ \frac{-i}{e^2}\star da + \frac{N}{2\pi}\varphi + \frac{\theta}{2\pi} \right] \right) \tag{59}$$

instead of Eq. (29), we would have run into two problems. First, the operator in Eq. (59) is not fully topological even without turning on a deformation due to the short-distance divergences discussed above. Second, this operator would completely fail to be topological once we add it to the action, since turning on the deformation would change the equation of motion for $a$, and the quantity in square brackets in Eq. (59) would no longer be constant. Neither of these problems appear when we use the correct expression for the LTO in Eq. (29).

As our last example, let us consider pure 4d Maxwell theory with action $S = \int_{M_4} \frac{1}{2e^2}\|da\|^2$. This theory has an "electric" $U(1)$ 1-form symmetry, and there is a widespread claim in the literature that the associated topological surface operators can be written as

$$U_\alpha(M_2) \stackrel{?}{=} \exp\left( i\alpha \int_{M_2} \star j \right) = \exp\left( i\alpha \int_{M_2} \frac{i}{e^2}\star da \right). \tag{60}$$

This expression suffers from the same issue as Eq. (51). To get a valid representation of $U_\alpha(M_2)$, we can integrate in a 2-form field $b^{(2)}$ and replace the Maxwell action by

$$S = \int_{M_4}\left[ \frac{e^2}{2}\left\| b^{(2)} \right\|^2 - ib^{(2)}\wedge da \right]. \tag{61}$$

If we eliminate the auxiliary field by using its equation of motion $b^{(2)} = \frac{i}{e^2}\star da$, we obtain the usual Maxwell action. So the symmetry operator for the electric $U(1)$ 1-form symmetry can be written as

$$U_\alpha(M_2) = \exp\left( i\alpha \int_{M_2} b^{(2)} \right). \tag{62}$$

It is now easy to check that $\langle U_\alpha(M_2)\rangle = 1$. We can also verify that with this definition of $U_\alpha(M_2)$ obeys the $U(1)$ fusion rule, as well as check that correlators of $U_\alpha(M_2)$ and Wilson loops satisfy the expected rule:

$$\left\langle U_\alpha(M_2)\exp\left( iq\oint_C a \right) \right\rangle = \frac{1}{Z}\int \mathcal{D}a\,\mathcal{D}b\, e^{-S}\exp\left( \int_{M_4}\left[ iq\,a\wedge\delta^{(3)}(C) + i\alpha\, b^{(2)}\wedge\delta^{(2)}(M_2) \right] \right)$$

$$= \exp\left( iq\alpha \int_{M_4}\delta^{(1)}(M_3)\wedge\delta^{(3)}(C) \right)\left\langle \exp\left( iq\oint_C a \right) \right\rangle, \tag{63}$$

where $\partial M_3 = M_2$, and $\int_{M_4}\delta^{(1)}(M_3)\wedge\delta^{(3)}(C)$ is the linking number of $M_2$ and $C$. To get to the second line, we changed variables in the path integral via the rule $a \to a + \alpha\,\delta^{(1)}(M_3)$. This completes the verification that Eq. (62) is a valid representation of the topological line operators that generate the electric $U(1)$ 1-form symmetry of 4d pure Maxwell theory.

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
