# Peer review of "Universal Deformations"

_SciPost Physics, doi:SciPost Phys. 12, 116 (2022)_

## Round 2 · Referee Report · Anonymous · 2021-12-6

Report

See the attached file.

Attachment

  • validity: -
  • significance: -
  • originality: -
  • clarity: -
  • formatting: -
  • grammar: -

Author:  Theo Jacobson  on 2021-12-09  [id 2020]

(in reply to Report 1 on 2021-12-06)
Category:
answer to question

We are very grateful to the referee for their helpful comments and questions, and
give some answers below:

The referee pointed out several typos - we have fixed them. Also, the referee
asked about what examples we had in mind other than the Schwinger model and pure
YM. There are many: 2d SU(N) adjoint QCD, 2d SU(2N) QCD with
two-index antisymmetric quarks, and so on.

(1) The referee asked for some comment on how the possibility of pair creation
is accounted for in our results. Consider for example the $N=2$ Schwinger model
with $m=0$, and an LTO deformation with a real coefficient $\Lambda^2$.

Let us place a test $q = 1$ charge at $x$, and a $q = -1$ charge at $x+L$, and
suppose that the charge at infinity vanishes. Equivalently, we can put a
charge-1 Wilson line running along the $y$ axis at $x$, and a charge-$-1$ Wilson
line running along the $y$ axis at $x+L$. Outside the region in between the two
Wilson lines we have the $k = 0$ universe, while in between the Wilson lines we have
the k=1 universe.

The difference in vacuum energies between the outside and inside can be read off
from our Eq. 2.17, and reads $|\Delta E| = \Lambda^2 | \cos(2\pi * 1/2) -
\cos(2\pi * 0/2)|$ (without $\Lambda$, the two universes would be degenerate
since $m=0$.) Now imagine a dynamical fermion-anti-fermion pair appears in
between the test charges. The dynamical fermions have charge $\pm 2$, so the
region in between the new pair of particles has the label $k=3$. Our analysis
shows that the vacuum energy density in the $k=3$ universe is the same as in the
$k=1$ universe, since the vacuum energies depend on $k$ through expressions like
$\cos(2\pi k/2)$.

All of our formulas in the Schwinger model are consistent with the statement
that the differences in vacuum energies are periodic in the universe labels in
e.g. a periodicity of N. One can interpret this periodicity in N as encoding the
possibility of Schwinger pair production.

(2) The referee also observed that, given the fact that the deformations we
consider only affect the vacuum energies of the universes, and the universes
can't talk to each other, it is natural (in the colloquial sense) that a quark mass
term is not generated. We agree - indeed this is exactly our argument in the
paper for the absence of a quark mass. The tension with the technical version
of naturalness comes about the following way. The discussion below already
appears in different words on e.g. page 17 in our paper.

For example, in the Schwinger model without a mass term or any other
deformations, there is a $\mathbb{Z}_N$ chiral symmetry. This chiral symmetry
makes the absence of a mass term in the interacting theory technically natural.
EFT naturalness says that if some term is forbidden by a global symmetry, and
then you break the symmetry by a relevant operator, then the forbidden term will
be generated by quantum fluctuations, with a scale determined by the scale of
the symmetry-breaking deformation. This principle is extremely widely used and
relied upon in QFT. And yet our analysis shows that it is violated in the
Schwinger model: one can break the $\mathbb{Z}_N$ symmetry by turning on a
(technically-relevant) ``universal deformation'', but this does not induce a
mass term, despite the fact that no symmetry protects it in the deformed
symmetry.

(The massless Schwinger model is very special, being essentially a free theory
in the massless limit, so it is worth emphasizing again that these remarks apply
very broadly to QFTs with the appropriate global symmetry, and the Schwinger
model is merely a helpful simple example.)

We hope that these remarks address the referee's questions, and that the paper is
ready to proceed to publication.

Anonymous on 2021-12-20  [id 2038]

(in reply to Theo Jacobson on 2021-12-09 [id 2020])
Category:
remark

The referee addressed all the issues in my report. I recommend this paper for publication.

---

## Round 2 · Referee Report · Anonymous · 2021-12-15

Strengths

1. The paper is clearly written
2. Section 5 has a good discussion on various conceptual points

Weaknesses

1. Some calculations could have been simplified and made shorter

Report

The authors discuss deformations of QFTs with topological local operators when they exist. Topological local operators lead to a decomposition of the theory into different "universes". The authors show that such relevant deformations correspond to adding different cosmological constant to each universe and have exactly calculable effects.

I recommend the draft for publication.

Requested changes

1. I suggest including a comment regarding the following issue:

An apparent distinction between invertible and non-invertible (d-1)-from symmetry has been made in the draft. However, since (d-1)-form symmetries are generated by local topological operators, one can always do a change of basis to use either invertible or non-invertible local topological operators as generators of the (d-1)-from symmetry. Hence there is no distinction between the two cases.

For instance in a unitary theory, there always exist the projection basis ($p_a$) of topological local operators satisfying the algebra:

$p_a p_b = \delta_{ab} p_a, \qquad a=1,\dots,N$

These are projection operators into different universes. Then one could use this basis to construct $O_m = \sum_{a=1}^N e^{2\pi i ma/N} p_a$ with invertible $\mathbb{Z}_N$ fusion rule.

Thus the fusion rule is not an invariant property of a (d-1)-form symmetry. Where as for a p-form symmetry with p<d-1, the fusion rule is invariant since a change of basis is not allowed for topological defects of higher dimension, e.g. line and surface defects.

  • validity: high
  • significance: good
  • originality: high
  • clarity: high
  • formatting: -
  • grammar: -

Author:  Theo Jacobson  on 2021-12-23  [id 2051]

(in reply to Report 2 on 2021-12-15)
Category:
remark

We're grateful to the referee for their helpful report!

We agree that given a set of invertible local topological operators, one can do
a change of basis to a set of non-invertible local topological operators.
This was emphasized in a recent paper by E. Sharpe we have already cited as our
Ref. 19.

However, we disagree with the claim that there is no invariant difference
between invertible and non-invertible 1-form symmetries in 2d. When one defines
the fusion rules obeyed by a set $S$ of operators $O_a$ (heuristically $O_a O_b
= \sum_{c} N_{a,b}^c O_c$), the set $S$ must be closed under the fusion, and we
think that is also natural to assume that it does not does not include the zero
operator. We have added a footnote about this on page 2 of our manuscript. The
fusion of two distinct projection operators is the zero operator, so this
assumption rules out using the `projection basis' when discussing fusion rules
and trying to sort out whether one is dealing with an invertible or
non-invertible symmetry. So we think it makes sense to draw a distinction
between symmetries generated by invertible and non-invertible operators,
especially in the context of our presentation.

Anonymous on 2021-12-23  [id 2053]

(in reply to Theo Jacobson on 2021-12-23 [id 2051])
Category:
remark

Thanks for the reply. In case that $O_a$ and $O_b$ are distinct projection operators, one concludes that $N_{a,b}^c=0$ for all basis element $O_c$ in $S$. This does not mean that zero has to be among the set $S$ which is a basis of the algebra. So it does not seem unnatural to use the projection basis.

If you have already added a footnote mentioning that you are choosing to work with a basis satisfying your criteria, then I have no further comment. As I mentioned before, I have no objection and recommend the draft for publication.

---

## Round 3 · List of Changes

-Fixed various typos and defined quantities that had been used without explanation.
-Added Footnote 1 to clarify a point raised by the second referee: we consider bases for local topological operators such that the zero operator does not appear in the fusion rules.
-Added Footnote 1 to clarify a point raised by the second referee: we consider bases for local topological operators such that the zero operator does not appear in the fusion rules.

---

## Editorial Decision

published